# Fusion Poser: 3D Human Pose Estimation Using Sparse IMUs and Head Trackers in Real Time

**DOI:** 10.3390/s22134846

**Published:** 2022-06-27

**Authors:** Meejin Kim, Sukwon Lee

**Affiliations:** Korea Electronics Technology Institute, Seongnam-si 13509, Gyeonggi-do, Korea; mj_kim1023@keti.re.kr

**Keywords:** IMU, human pose estimation, real time, motion reconstruction, sensor fusion, inertial sensors

## Abstract

The motion capture method using sparse inertial sensors is an approach for solving the occlusion and economic problems in vision-based methods, which is suitable for virtual reality applications and works in complex environments. However, VR applications need to track the location of the user in real-world space, which is hard to obtain using only inertial sensors. In this paper, we present Fusion Poser, which combines the deep learning-based pose estimation and location tracking method with six inertial measurement units and a head tracking sensor that provides head-mounted displays. To estimate human poses, we propose a bidirectional recurrent neural network with a convolutional long short-term memory layer that achieves higher accuracy and stability by preserving spatio-temporal properties. To locate a user with real-world coordinates, our method integrates the results of an estimated joint pose with the pose of the tracker. To train the model, we gathered public motion capture datasets of synthesized IMU measurement data, as well as creating a real-world dataset. In the evaluation, our method showed higher accuracy and a more robust estimation performance, especially when the user adopted lower poses, such as a squat or a bow.

## 1. Introduction

Reconstructing human poses with a 3D skeleton-based body model and recording the associated motions, commonly called motion capture (mocap), has significant influences on computer vision, animation, robotics, and biomechanics. For example, many fields use captured motion to create better visual effects, such as in games and movies. It is also used to analyze a subject’s movement for medical or military purposes. Nowadays, there are demands for motion capture in virtual reality (VR) applications to interact with virtual objects in real time.

Typically, commercial full-body motion capture systems are optical, such as Vicon [1] and OptiTrack [2]. When a user moves while wearing a suit that is covered with markers, the motion is estimated by analyzing the 3D positions of the markers, which are projected on multiple calibrated cameras. This marker-based system can restrict space and mobility, depending on the installation of the cameras. Frequently, dedicated studios are carefully designed using pre-installed cameras to prevent occlusions and lighting disturbances. Furthermore, vision-based methods are affected by reflections and illuminations, which make it difficult to record motion outdoors. Despite these constraints, marker-based methods achieve high-quality results. However, depending on the financial and computational costs and installation difficulties, vision researchers have focused on obtaining pose estimation using a few RGB [3,4,5] or RGB-D [6,7,8] images. These methods are much simpler but allow for limited space due to the camera. Another approach uses finely controlled moving cameras [9,10,11], but its usability is still limited.

Another motion capture system uses multiple inertial measurement units (IMUs) that estimate the orientation by combining a gyroscope, magnetometer, and accelerometer. In contrast to optical systems, motion capture using wearable inertial sensors is less affected by environmental constraints and occlusion. In addition, it does not require a complex facility setup or expensive costs, making it suitable for individual users in VR.

Xsens [12] and Perception Neuron [13], which are representative commercial inertial motion capture systems, estimate joint parameters using 17 IMUs. The equipment requirements of these systems decrease accessibility and make them unsuitable for personal use. Recently, von Marcard et al. [14] presented a method for full-body pose estimation using six IMUs. However, this study requires offline optimization, which is computationally expensive. In addition, Deep Inertial Poser [15] was the first study to apply a deep learning method to 3D pose estimation using six IMUs in real time. It proposed a bidirectional recurrent neural network (BiRNN) model, which was trained using captured and synthesized datasets. However, this method cannot estimate the position of the full body, only the local orientation of the joints.

Head-mounted displays (HMDs), which are widely used in VR fields, can track the position and rotation of the user’s head with respect to the environment. By combining the head position with data from IMU sensors, our method can improve the pose estimation quality because the head’s height provides more information than just data from IMUs. In addition, the hip position can be estimated from the position of the HMD. Because of these points, our method can robustly estimate full-body poses and its position within the virtual space.

This paper proposes a human pose estimation method that uses inertial sensors and a VR HMD that provides the positional information. We had three challenges to overcome for our method to be suitable for VR: (1) it had to be operated in real time; (2) the errors in the joint angles or distances had to be minimized; and (3) it had to track the global position of the user at the same time. To this end, we introduced a network that combines biRNN and convolutional long short-term memory (convLSTM) [16] layers to deal with these challenges using six IMUs and a head tracker. This network addresses the following constraints: (1) unlike optimization approaches, the learning-based method needs a relatively shorter time to produce the prediction, which could meet the real-time constraints; (2) because of the continuity of human motion, future movements depend on the current and past movements, so the suggested network improves accuracy and continuity by learning spatio-temporal properties from the datasets (despite an insufficient number of sensors); (3) because of the accuracy, the human poses in the training dataset are presented with body-centric coordinates, which are described Section 3.2.1, but they only show the pose of the user, not the location or the direction in which the user is looking. Out method merges the head pose from the HMD with the predicted pose, which is local information, to recover global information. Furthermore, the hip velocity, which is one of the network outputs, makes the hip trajectory more continuous.

Our proposed network is divided into 2 phases: joint position estimation and joint rotation estimation, as referred to as P1 and P2 in Section 3.2.2, respectively. The joint rotation is predicted after the joint position because the rotation estimation network takes the predicted position as its input recursively. The IMU sensors measure the angular velocity, acceleration, and magnetic field and calculate the user’s orientation using the sensor fusion algorithm. Our method takes the orientation and acceleration data from the IMUs as the inputs for the position estimation network. In contrast to other inertial-based estimation methods, our approach also uses head height data from the head tracker, which is measured as the distance from the ground to the height of the HMD. This measurement makes the prediction more robust since it removes ambiguity by providing global information; for example, when the placement of the sensor is on the lower leg, the orientation of the sensor is the same in standing and sitting postures.

We acquired human motion data using OptiTrack [2] and Xsens DOT [12] to create a dataset to train the model. To obtain more datasets, we synthesized full-body joint data and the orientation and acceleration of inertial measurements from open datasets, such as the CMU 3D motion dataset [17] and TotalCapture [18].

We evaluated our model using effectiveness comparisons for network model variances and input types and using comparisons to other works, such as DIP [15]. We used two evaluation data metrics that are widely used in pose estimation studies: the mean per joint position error (MPJPE) and the joint angle error. The evaluation results showed that the proposed model could obtain the global position with a higher pose estimation accuracy. We also implemented real-time applications to show that our model could be applied to VR.

## 2. Related Work

Our proposed pose estimation model uses sparse inertial sensors and a head tracking sensor to determine human movement and full-body posture. Motion capture methods can be classified according to their input parameters, including method that use multiple sensors. Here, we introduce a related work analysis of vision-based methods that use cameras or markers, methods that only use inertial sensors, and methods for integrating signals from the sensors.

### 2.1. Vision-Based Motion Capture Methods

Vision-based motion capture, which is the classic method for obtaining human motion, has been the focus of various studies throughout its long history. In particular, commercial motion capture uses a large number of markers and multiple calibrated cameras. Several studies [19,20,21,22,23,24] have made efforts to overcome the shortcomings of the popular approaches that use single or multiple cameras. Many methods that require high estimation accuracy are conducted offline [25,26,27,28,29,30,31,32,33]. Recently, real-time studies have also been proposed. VNect [4] is a representative study on 3D kinematic human pose estimation in real time (30 Hz), which combines fully convolutional neural networks. As in previous studies, deep learning techniques have significantly improved the pose estimation method. Since DeepPose [3] was proposed, which was the first major 2D human pose estimation study to apply deep neural networks (DNNs), convolutional networks (ConvNets) that are based markerless motion capture analysis [34,35,36,37,38,39] have been generalized. In addition, other studies have used multi-view images [40,41] or single depth images to obtain high accuracy [6,7,8]. In contrast to vision-based methods, our approach uses a system that can be installed without significant restrictions. In order to overcome the problems with vision-based methods, such as occlusion (depending on where the cameras are installed), we combined sensor systems that are not highly affected by direction.

### 2.2. Full-Body Sensor-Based Motion Capture Methods

Methods that use inertial sensors are another broadly used approach to commercial motion capture. Typically, Xsens MVN [42] conducts six degrees of freedom (DOF) full-body motion tracking using 17 IMU sensors that take measurements from a combination of accelerometers, gyroscopes, and magnetometers. Compared to vision-based methods, IMU motion capture is easier to use in out-of-lab situations as it reduces spatial constraints. However, the large number of inertial sensors that is required has the problem of high costs and being time-consuming to set up. Therefore, existing studies have tried using a small number of sensors, despite the performance degradation. Some studies [43,44] have constructed human poses using only five accelerometers by retrieving pre-recorded poses with similar accelerations from a motion capture database. In these studies, the measurement instability of the sensors and the size of the database excessively affected the performance of the method. Recently, research has been conducted on reducing the number of sensors by using inertial sensors that can measure acceleration and orientation simultaneously. A pioneering work in this field, Sparse Inertial Poser (SIP) [14], presented a joint optimization model that reconstructs the pose of SMPL body model [45] using six IMUs but without relying on databases. To advance SIP, Deep Inertial Poser (DIP) [15] adopted a deep learning method for running in real time. DIP uses a BiRNN [46] with LSTM cells [47]. This approach has the potential for real-time 3D pose estimation in VR environments, which provided us with great motivation. However, DIP cannot estimate the global movement of the user, which is an imperative component of tracking motion. Yi et al. [48] proposed TransPose to estimate global translations by using a supporting foot-based method and an RNN-based method. TransPose achieves a state-of-the-art performance in terms of pose estimation accuracy using only six IMUs. Our proposed model estimates the position of each joint and uses the 3D position of the head (which is obtained from the head tracking sensor) to increase the accuracy of the motion tracking. In addition, by using the head position, our study achieved real-time and full-body human motion estimation within real-world space by obtaining human movements that play important roles within VR.

### 2.3. Performance Optimization Based on IMUs

Reconstructing human poses from sparse IMUs to a high degree of accuracy is a challenging problem because the data from the sensors are insufficient for configuring human poses. Many researchers have studied the sensor fusion method using inertial sensors along with other sensors or cameras to increase the estimation quality. Some studies [49,50] have applied six inertial and ultrasonic sensors to obtain 3D positions and orientations. Liu et al. [49] proposed a method for online pose estimation that retrieves data with similar signal configurations from pre-defined motion databases. Another approach is to combine inertial sensors with videos [51,52,53], especially multi-viewpoint videos (MVVs) [18,54,55,56], depth cameras [57,58] or optimal markers [59]. Total Capture [18] fuses MVVs with inertial measurement units and applies a convolutional neural network (CNN) output layer to an LSTM model. The use of the sensor fusion model with cameras significantly increases estimation accuracy but still includes several challenges, such as occlusion, lighting problems, installation complexity, and limitations in mobility. Our proposed model is another sensor fusion model, which combines the signals from inertial sensors with signals from a head tracker. The head tracker records the 3D position of the head, is consciously used in VR to track the location of the HMD, and has a positive effect on estimating the global position and full-body pose of a human in real-world space. Therefore, our method increases the accuracy of human pose estimation while using fewer sensors compared to existing studies and facilitates its application in virtual environments.

## 3. Method

The proposed method estimates the full-body pose and pelvis position of the user via an HMD and IMU bands. A biRNN network with a ConvLSTM [16] layer is then used to predict joint position and rotation, which takes the orientation and acceleration sequences from the IMUs as its input. In addition, we integrate the estimated joint pose with the head pose from the HMD to calculate the pelvis position. In this paper, we first introduce our approach in Section 3.1. We explain the structure of the network in Section 3.2. We then describe the method for reconstructing global positions from local information in Section 3.3.

### 3.1. An Overview

Our approach requires two types of sensors: IMUs and global head trackers. Six IMU sensors that are placed on pre-specified body parts are used to predict the joint position and orientation of the user. Moreover, it is an underdetermined method because the number of input data is relatively sparse compared to the number of joints that need to be estimated. Thus, we developed a pose estimation network (Section 3.2) that predicts full-body poses based on training data from the measurements of sparse IMUs. We defined body-centric coordinates (Section 3.2.1) that describe every joint in the frame that is located on the root joint for learning efficiency and consistency. Using the body-centric coordinates, we can remove all global information, such as the global position and orientation, except for the height of the head, which changes with every movement. When predicting the pose as an output, the removed global information needs to be recovered to place the user within the virtual space. To this end, our approach uses an additional sensor: a global head tracker that can be an HMD or a motion capture device. In addition, the pose estimation network takes head height as one of its inputs. The head height removes any ambiguity that comes from the sparse IMUs. At the end of the procedure, the tracked head pose from the HMD is combined with the predicted human pose from the network by locating the head pose of the HMD. When two head poses are identified without other processes, there could be a foot sliding problem: the foot could move in the air because of the incompetence of the prediction. To solve this problem, we introduced the velocity term at the network output to constrain the movement of the root joint. As another solution, we could use the velocity of the IMU that was located on the root joint, but a drift occurred as time passed. Thus, the predicted root joint velocity is used by averaging it with the velocity from the HMD. Figure 1 illustrates the entire process of our method and the more detailed parts are explained in the next section with formal definitions.

### 3.2. The Pose Estimation Network

We defined the input and output of the network at time *t* as $X_{t}$ and $Y_{t}$, respectively. The input $X_{t}$ is the sequence of the sensor measurement data $x_{f}$, where *f* is a neighbor frame of the time *t*. $X_{t}$ and $x_{t}$ are as follows: (1)$$X_{t}=\left[x_{f+t}|f=\left\{-15,\;-10,\;-7,\;-4,\;-2,\;-1,\;0,\;2,\;4,\;6\right\}\right],$$
(2)$$x_{t}=\left[y^{head},\;{\hat{a}}^{limbs},\;{\hat{q}}^{limbs}\right]\in R^{43},$$
where $y^{head}\in R$ is the height of the head (as measured by the tracker), ${\hat{a}}^{limbs}$ and ${\hat{q}}^{limbs}$ represent the acceleration and quaternion of the IMU on each limbs. For more clarity, each of the mathematical forms of ${\hat{a}}^{limbs}$ and ${\hat{q}}^{limbs}$ was defined as: (3)$${\hat{a}}^{limbs}=\left[a_{x}^{l},a_{y}^{l},a_{z}^{l}\right]\in R^{18}\;\mathrm{and}\;{\hat{q}}^{limbs}=\left[q_{x}^{l},q_{y}^{l},q_{z}^{l},q_{w}^{l}\right]\in R^{24},$$
where *l* is the list of limbs on which the IMU sensors are placed. In our experiment, the sensors were placed on the right hand, left hand, pelvis, head, right foot, and left foot, (cf. Figure 5). Next, we defined $Y_{t}$, which is the output of the network, as follows: (4)$$Y_{t}=\left[p^{joints},\;q^{joints},\;v_{xz}^{root}\right],$$
where $p^{joints}$ is the concatenated positions of full-body joints in the motion data and, similarly, $q^{joints}$ is the concatenated quaternions. Lastly, $v_{xz}^{root}\in R^{2}$ is the velocity of the root joint with respect to the x,z plane.

#### 3.2.1. The Body-Centric Coordinates

As mentioned above, $Y_{t}$ is described in the body-centric coordinates as the frame that is placed on the root joint. Because the root joint describes every joint and only has a global position and orientation, every joint loses its global information when the user moves and aligns the root joint with the origin. When the global data are not removed, $Y_{t}$ can have different values, even when the user adopts the same poses in slightly different locations. We defined the local frame as follows:(5)$$T\left(t\right)=[R_{t}^{root}\left(\theta_{y}\right);p_{t}^{root}].$$

As the origin of the local frame is placed at the root joint position, $p_{root}$, the global translation can be removed. In addition, to remove the global orientation, the z-axis of the local frame $R^{root}\left(\theta_{y}\right)$ has the same direction as the z-axis of the root joint, which is the forward direction. Because the local frame is rotated about the y-axis, the rotation of the x- and z-axis can be preserved, which is essential for dealing with a more realistic pose; for example, a bowing or running pose requires the x-axis rotation of the root joint with respect to the real-world space. T depends on time *t*, so computing T(t) is described in Section 3.3.

#### 3.2.2. Network Architecture

Using these definitions, we introduced a pose estimation network that predicts a single pose that corresponds to the input X at the current frame. Deriving high-accuracy poses from sparse acceleration and orientation data is a challenging task. Our proposed model focuses on two challenging solutions: (1) the naturalness of the motion and (2) the constraints of the human body structure. In a previous study on DIP [15], a biRNN [46] with LSTM [47] cells was proposed, which is suitable for use with time series learning to predict the SMPL pose parameters from IMU inputs. The biRNN model can access frames in two directions (past and future) and maintain the temporal and structural properties of motion in natural movements. Inspired by this work, we adopted the biRNN model pose reconstruction. However, the output of our method is directly composed of the joint position and rotation, which makes the usage of the output simpler, and the structural constraints are maintained well without SMPL. To reconstruct a pose, the rotational data $q^{joints}$ must be assigned to every joint. Before estimating the rotation, the positional data $p^{joints}$ are predicted using the measurement data from the ${\hat{a}}^{limbs}$ and ${\hat{q}}^{limbs}$ IMUs and the head height $y^{head}$, which then become the input for prediction of the rotation $q^{joints}$. Thus, the network architecture has a two-stage structure: the first stage infers the positional aspect of the pose; the second stage infers the rotational aspect by using the output of the first stage to predict the rotational pose of the IMU sensors.

We describe the architecture of the pose estimation network in Figure 2.

First of all, the input $X_{t}$ is divided and rearranged according to its meaning, instead of the time sequence. As the result, the measurement data are reshaped into a four-dimensional matrix, of which each dimension is $\left(\#\mathrm{frames}\times \#\mathrm{IMUs}\times \left(\hat{a},\hat{q}\right)\in R^{7}\times 1\right)$. After reshaping, the output is fed into the convolutional LSTM layer to perform spatio-temporal learning, which enables the network to learn the relationship between the sensor data and time more effectively. In addition, this method shows a good ability to preserve pose stability rather than using the measurement data directly. The output of the convolutional LSTM is concatenated with the current frame $x_{t}$ and the sequence of the head height data $y^{head}$, which is then used as the input for the bidirectional LSTM layer. The biRNN [46] and long short-term memory (LSTM) [47] cells are used to compute the optimal weights through continuous sequence learning. Since our method uses a sparse number of IMU sensors in relation to the size of the human body, the sensors have to provide sufficient information to generate full-body poses. Although frame-to-frame changes are applied using the acceleration values, as mentioned in the DIP study [15], the acceleration has less of an influence on the predicted results than the orientation. Therefore, we use bidirectional LSTM layers because of the continuity of the motion. We also needed to consider future consequences and distinguish between actions that have the same orientation and acceleration values as IMUs. The output of the joint position estimation phase was defined $Y_{t}^{P1}$ as:(6)$$Y^{P1}=\left[p^{joints},\;v_{xz}^{root}\right].$$

In the second phase, the subnetwork P2 in Figure 2 predicts the joint rotation ${\hat{q}}^{joints}$ of the output $Y_{t}$ based on two inputs: the rotation data from the $q^{limbs}$ measurements and the results of the subnetwork $Y^{P1}$. The subnetwork P2 consists of unidirectional RNNs with LSTM cells. We determined that the joint position data that are estimated in P1 provide enough information to reconstruct the rotational information $q^{joints}$. Note that, unlike the subnetwork, P1 requires a sequence of the frames, whereas P2 depends on the current frame *t*.

### 3.3. Reconstructing Global Poses

Because the network output $Y_{t}$ is with respect to the body-centric coordinates, the HMD position is combined with the output to reconstruct the root trajectory. To this end, the local frame T(t) (Equation (Equation 5)) needs to be computed, but we could not obtain exact values for $[R_{t}^{root}$ and $p_{t}^{root}$ because the positions of the sensors differ every time the user wears them. We introduced the calibration step to complete the parameters of T(t), during which the user aligns the directions of the head and root; for example, the A-pose or T-pose. Firstly, we assumed that the z-axis direction of the tracker would coincide with the facing direction, the $q_{head}^{w}$ of the IMU sensor would be represented as real-world coordinates, and that it can be easily satisfied by stacking the head tracker with an IMU sensor. Using this setup, the rotation aspect at calibration time *c* can be calculated as follows:(7)$$\phi_{y}^{*}\leftarrow \arg\min_{\phi_{y}}\left|\right|R_{c}^{tracker}z-R\left(\phi_{y}\right){\hat{R}}_{c}^{head}z{\left|\right|}^{2}$$
where $R^{tracker}$ and ${\hat{R}}^{head}$ are the rotation matrices of the tracker and IMU sensor, respectively, and z is a unit vector [0,0,1]. After $R(\phi_{y}^{*})$ is multiplied by ${\hat{R}}^{head}$, the projection of the transformed z has the same direction as one of the trackers. The rotation matrix of the root at time *t* can then be calculated by applying the results of Equation (Equation 7) to the measurement data from the IMU sensors:(8)$$R_{t}^{root}={\hat{R}}_{t}^{root}\;\cdot \;{\left({\hat{R}}_{c}^{root}\right)}^{-1}\;\cdot \;R\left(\phi_{y}^{*}\right)\;\cdot \;{\hat{R}}_{c}^{root}$$

In this equation, ${\hat{R}}_{c}^{root}$ is the rotation matrix of the measurement data from the root joint at the calibration time *c* and thus, ${\hat{R}}_{c}^{root}$ and $\theta_{y}^{*}$ are stored to obtain $R_{t}^{root}$. To predict the rotational aspect of the transformation *T*, only the angle that rotates about the y-axis is needed, which can be solved in a similar way to Equation (Equation 7):(9)$$\theta_{y}^{*}\leftarrow \arg\min_{\theta_{y}}\left|\right|R_{t}^{root}z-R\left(\theta_{y}\right)z{\left|\right|}^{2}$$

After computing the rotational aspect of parameter of *T*, $p_{root}$ can be obtained by applying the position of the head from $Y_{t}$ (see Figure 3):(10)$$p_{t}^{root}=p_{t}^{tracker}-R\left(\theta_{y}^{*}\right)p_{t}^{head}$$

Because the root position follows the position of the head, the quality of the root trajectory depends on the quality of the estimation. However, the noise in the estimation cannot be removed. As a result, the root trajectory shows an unwanted jerk that lowers the motion quality. To solve this problem, we introduced the velocity term $v_{xz}^{root}$ to the output $Y_{t}$, which constrains the velocity of the root joint using a simple weighted average:(11)$$p_{t}^{xz^{\prime }}=p_{t-1}^{xz}+\alpha \left(p_{t}^{xz}-p_{t-1}^{xz}\right)+\left(1-\alpha \right)v_{xz}^{root}$$

For simplicity, the superscript of *p* is omitted in the above equation, which is related to the root joint. In our experiments, the value of 0.1 for α worked well, which meant that the results depended more on the prediction.

## 4. Datasets

This section describes the configuration of the datasets that were used for the model implementation in more detail. We introduce the skeleton structure that constructs the human pose data in Section 4.1, the detailed instructions for the motion capture and IMU data in Section 4.2, the method for IMU calibration in Section 4.3, and the synthetic data in Section 4.4.

### 4.1. Skeleton Structure

The joint position $p^{joints}$ provides the position of the human body joints using the body-centric coordinates. Figure 4 depicts a skeleton that consists of 21 joints ($p^{1}$ to $p^{21}$). The height of the avatar configuration that is set during motion capture is the size of the skeleton structure, within which the joint positions are determined in centimeters (cm). The joint placement of our skeleton structure was based on the full-body motion capture data that we collected.

### 4.2. Motion Capture and IMUs

We utilize two types of sensor data for our data-driven model: motion capture data and IMU data. In the experiments, we recorded raw data from a subject who was wearing an OptiTrack [2] motion suit with 50 markers and 6 Xsens IMU sensors. As shown in Figure 5, six IMUs were mounted on the pelvis, the left and right hands, the left and right legs, and the head. The subject executed the calibration steps for the optical markers on the suit. After calibration, the participant performed actions following pre-defined scripts, such as locomotion, sitting, crawling, and other motions. The ground-truth motion capture data were recorded in 120 Hz and IMU data were recorded in 30 Hz, so data synchronization problems could occur due to the frequency differences. Therefore, we applied linear interpolation according to the timestamps of the two sets of data.

### 4.3. Sensor Calibration

We used Xsens DOT IMU sensors, which contain 3-axis accelerometers, gyroscopes, and magnetometers. The measurement of the IMUs was represented using the local coordinate system, which was defined as right-handed Cartesian coordinates, and thus, each IMU had different coordinates. To obtain the identified coordinate system between the sensors, we used the heading reset function on the IMU sensors that aligns magnetic north with the forward direction of the physical body. After the calibration step, we could obtain the measurement data from the six IMUs in terms of the inertial coordinate system. For training, we converted the measurement data into the body-centric coordinate system, which is described in Section 3.2.1. On the other hand, we used the inertial coordinate system for the predictions.

### 4.4. Generating Synthetic Data

To perform the predictions, the network requires a large amount of data, but when only the data from the motion capture are used, the cost of the data is unaffordable. To this end, many works [15,48] have generated synthetic data from existing motion capture datasets by simulating the measurement data from the IMUs, which is the method that we adopted to carry out the predictions. We generated synthetic data using the CMU 3D motion dataset [17] and TotalCapture [18] and uncommon behavior motions were excluded, such as sport, dance, and martial arts. To simulate the measurement data from the IMUs, we used the following steps: (1) we retargeted the motions from datasets onto our skeleton (Section 4.1) for consistency; (2) we placed virtual IMUs on the skeleton where the physical IMU sensors were placed; (3) lastly, we calculated the orientation $\hat{q}$ and acceleration $\hat{a}$ by synthesizing the motions of the virtual IMUs followed by smoothing with the B-spline curve to obtain the motion trajectory.

## 5. Experiments

Before this proposal, we conducted experiments to carry out a quantitative and qualitative evaluation of our pose estimation model. This section summarizes our experiments. First, we introduce the data and metrics that we used for the experiments in Section 5.1. We evaluate and compare the performance of our pose prediction model using real-time settings in Section 5.2. In Section 5.3, we introduce the implementation of a real-time application using our pose estimation network and global location tracking method. Finally, we describe the hardware settings for our work in Section 5.4.

### 5.1. Data and Metrics

The experiments were conducted using the real TotalCapture [18] dataset. The validation dataset consisted of 10 consecutive input frame sequences that were not used for training. We used the mean per joint position error (MPJPE), joint angle error, and location tracking error as the metrics for the quantitative evaluation. First, we calculated the mean value of the Euclidean distance between the expected position and the obtained position for 15 major joints that were representing a real-time pose. Then, we calculated the mean error per joint from the difference in degrees between the predicted and the actual movement and the difference in root distance between the reconstructed global position and the recorded position in real-world space.

### 5.2. Evaluation

#### 5.2.1. Quantitative Evaluation

We evaluated the following variants to identify the model configuration that produced the best performance: (1) estimations with different components at the input and (2) estimations from reconstructing the network architecture.

*Measuring the errors for comparison.* To measure the positional errors of DIP and TransPose, we reconstructed an SMPL mesh model and the joint positions from the outputs of these works, which provided the SMPL parameters. To compare those results to ours, we used the same joints to measure the errors, as defined in Figure 4.

*Influence of input components*. To compare the effectiveness of the different input types, we experimented with three different types of networks: one network only used the current time *t* (only current), one network did not use the head height (non-head), and the other network did not use the acceleration of the IMUs (non-acc). Table 1 shows the results of these experiments. As the table shows, the positional error of the “only current” network was higher than that of “Ours”, which used ten sequences. Moreover, when measuring the lower body error, the mean error of “Ours” was 49.18 mm (± 29.50 mm) and that of the “only current” was 52.65 mm (± 29.16 mm). This indicated that using past-to-future information led to an improved pose estimation performance. Furthermore, as shown in Figure 6, the model without head height data showed a significantly higher position error 55.45 mm (±27.19 mm). This result was because this model could not track extreme changes in full-body poses, such as bending over or sitting on the floor (cf. Figure 7). It can also be seen that the acceleration data from the inertial sensors improved the joint position estimation accuracy by including the relative position differences in the input layer. Therefore, we identified a solution for estimating a wider range of poses than previous studies by adding head height data to the input using a head tracker.

*Influence of network architecture.* The proposed network consists of a convolutional LSTM layer and bidirectional LSTM layers. These RNN layers were added to extend the capability of predicting more accurate full-body poses. In this evaluation, we compared three network variants: (1) a network consisting of a fully connected layer (FC); (2) a network using a unidirectional RNN layer, not bidirectional (Conv+RNN); and (3) a network with no convolution layers (biRNN). Figure 6 shows the influence of the network structure on positional errors using the validation dataset. The distributions of positional errors using the three tested cases indicated a higher error frequency than that using our network structure. Further, Table 1 shows that the joint angle errors in these cases had slightly lower values. However, the experimental results showed that our configuration would be more suitable for real-time application as it showed a higher performance for continuous frames.

*Location tracking error.*Figure 8 shows the root trajectory that was estimated using our method and that of the ground-truth. The mean location error was 9.4148 cm (±7.9058). For the evaluation, we used the walking motions of Subject 4 in the TotalCapture dataset because it was suitable for demonstrating the root trajectory. Figure 9 shows the discrepancy between the full-body poses by overlaying the ground-truth and reconstructed results. It shows that the reconstructed path achieved the intended result with high accuracy.

#### 5.2.2. Qualitative Evaluation

As described in Section 5.2.1, we experimentally determined the best performance configuration. This section presents the poses that were estimated using our network as 3D body models and compares these poses to the ground-truth. We provided this comparison using our mocap dataset and the TotalCapture dataset [18]. In this section, we also describe the differences between the poses in terms of the evaluation variants that were difficult to express numerically.

Figure 7 shows the differences between two models that were trained in different ways. The first model was trained without synthetic data and the second model was the network variant that did not use the head height data. This figure shows that the proposed approach was more promising than the other variants. Moreover, in full-body exercises such as stretching or jumping, the height of the head provides reliable information for distinguishing poses and can be easily acquired from HMDs.

*Comparison to ground-truth poses.*Figure 10 shows some example prediction results using different poses from our mocap dataset and the TotalCapture dataset [18]. The ground-truth (GT) pose on the left was captured using a large number of optical markers from both datasets and the pose on the right was estimated with our method using six IMUs and one head tracker. Although there were challenging issues, such as hand pose (as detailed in Section 6.1), we could use our method for human pose estimation in real time.

*Comparison to previous works.*Figure 11 shows a visual comparison between the online pose estimation results of our method and those of other state-of-the-art works. The figure shows a qualitative comparison between our predicted poses and the SMPL poses that were estimated using DIP and TransPose for example frames from the TotalCapture dataset. Our approach estimated more accurate results for relative joint positions in the upper and lower body.

### 5.3. Real-Time Application

In this study, we implemented a real-time application using a VR HMD, as described in Figure 12. Our application received the measurement data from six IMU sensors and continuously estimated the user’s pose. At the same time, the pose of the head served as the input and global information for reconstructing the root trajectory. Our real-time application ran in the Unity 3D environment. Since the biLSTM layer required future data from the predicted time *t*, we stacked 10 sequences before the prediction, causing the application to have a delay of around 0.3 s. Because of the delay, there could be a discrepancy between the head position and the predicted pose when the user moved relatively fast.

### 5.4. Hardware Configurations

We trained our model using an Intel(R) Core(TM) i9-10900K CPU and an NVIDIA RTX 3090 graphics card. The real-time application ran on another PC with an Intel i5-10600 CPU and an NVIDIA RTX 2060 graphics card. We used Xsens [12] DOT IMU sensors to record the IMU data, both for training and real-time data. The ground-truth motion data were captured using an OptiTrack [2] Prime Camera and a motion suit with 50 markers. In addition, we used the Antilatency [60] tracking system to track the head position in real-world space.

## 6. Discussion

This paper introduced a pose estimation method that uses six inertial sensors and a head tracker to reconstruct human poses and global body positions in real time. Our model has the following novelties: (1) an improvement in human pose estimation by adding head position data; (2) the provision of a reliable global position, which is essential for VR applications; and (3) the acquisition of a higher accuracy for pose estimation by combining spatio-temporal layers and body-centric coordinates. We showed better quantitative results using the head position data and model configuration (cf. Section 5.2). Moreover, although we adopted an economically efficient type of sensor, the method had fewer restrictions on action and mobility. Nevertheless, the noise accumulation problem over time when using IMUs and some other limitations remain a challenge (Figure 13).

### 6.1. Limitations

The motion capture dataset that was collected to train our model included various actions, but it could not respond to extreme changes in the position of the pelvis, for example, when crawling or lying down. When the waist and the floor were parallel, as shown in Figure 14, the predicted pose and the body rotation were not similar to the ground-truth. We posit that the pelvis rotation caused the pose errors as the IMU data that were used for training were transformed into body-centric coordinates.

It is also challenging to determine hand poses when using a small amount of IMU data. In this paper, the only data that could determine the hand poses were from a pair of sensors that were worn on the wrists, but these were insufficient data to track the wrist rotation. The right-hand side of Figure 15 shows an example of different hand poses for which the wrist rotation was not predicted correctly during the motion of putting the hands together.

We extended the dataset by adding synthetic data to our mocap data to improve the performance of our network (cf. Figure 7). We built a 3D body model to simulate the measurement data from the IMUs and generate the output Y. We manually retargeted all of the datasets for prediction accuracy, but the retarget process was carried out per subject, which required a small amount of labor compared to what would be required for a whole dataset. However, we observed that the distortion of the retargeted body was due to the limitations of the retargeting method. For example, on the left-hand of Figure 15, the problem of the shape twisting according to the movement of the body can be seen. Although this distortion was seen in a low proportion in the overall data, it could be analyzed as the cause of low accuracy for specific poses.

## 7. Conclusions

In this paper, we introduced Fusion Poser, which estimates the pose of a user who is wearing six IMUs and translates the world coordinates of a head tracker in real time. The orientation and acceleration of the inertial sensors and the head height data are used as network inputs to estimate joint position, joint rotation, and root velocity. Our network architecture mainly adopts biLSTM layers to maintain the spatio-temporal relationship between the joints. The convLSTM layer is applied to the IMU sequence before the biLSTM layer to improve the prediction quality. In addition, the LSTM layer shows higher accuracy for estimating the orientation of a joint than the fully connected layer, as shown in Table 1. This method requires a large dataset to train the proposed network, which is cost-intensive to gather using motion capture. For cost-effectiveness, synthesized data can also be used by simulating the measurement data of virtual IMUs and models from open datasets, such as CMU and TotalCapture. For pre-processing, the coordinates of the output Y are converted into body-centric coordinates, which enables effective learning by removing global information. For the estimations, the translation and orientation of the root joint are recovered using the head tracker. In our experiments using the TotalCapture dataset, our method achieved a mean per joint position error of about 50 mm and a mean per joint angle error of about 11.31°, which was a better performance than those of the compared works [15]. Our approach requires a head sensor to track the pose, but this is commonly implemented using HMDs.

## Figures and Tables

**Figure 1 sensors-22-04846-f001:**
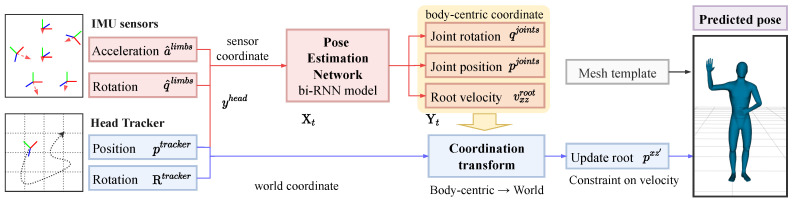
An overview of the proposed method. Our method uses two types of sensors to estimate human poses. IMU sensors are attached to each limb to measure their inertial data (orientation and acceleration), which are then input into the pose estimation network along with the head height to predict the user’s joint position and rotation. By combining global head poses from the head tracker with the joint positions, our method can predict the global full-body pose of the user.

**Figure 2 sensors-22-04846-f002:**
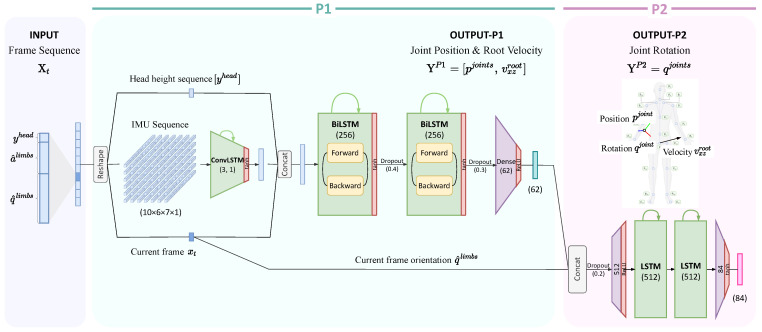
The Fusion Poser network architecture. The model inputs are the sequence of IMU sensor data and the height of the head. The length of the sequence is 10 time intervals, each of which has 43 features. The network consists of two stages: Phase 1 (P1) predicts the joint position using the IMU data and the head height sequences with the biLSTM layers, followed by the 4D convLSTM layers; Phase 2 predicts the joint orientation at the current time *t* using the output of P1 and the head height sequences with the LSTM layers.

**Figure 3 sensors-22-04846-f003:**
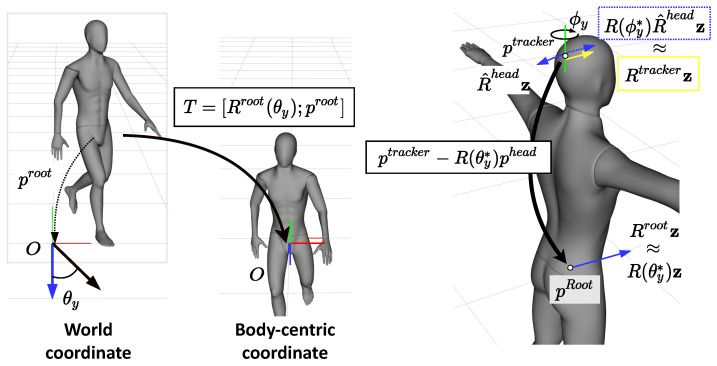
The coordination transformation: (**Left**) for training, we removed global information by transforming the real-world coordinates into the body-centric coordinates that identified the position of the frame at the root joint and the rotation about the y-axis was the direction of the root joint aligning to the z-axis; (**Right**) after the prediction, the global information had to be recovered, so the rotation $\phi_{y}^{*}$ was computed by matching the tracker’s direction with the IMU that was attached to the head. With rotation $R(\theta_{y}^{*})$, the position of the root joint could be computed from the head position using our network.

**Figure 4 sensors-22-04846-f004:**
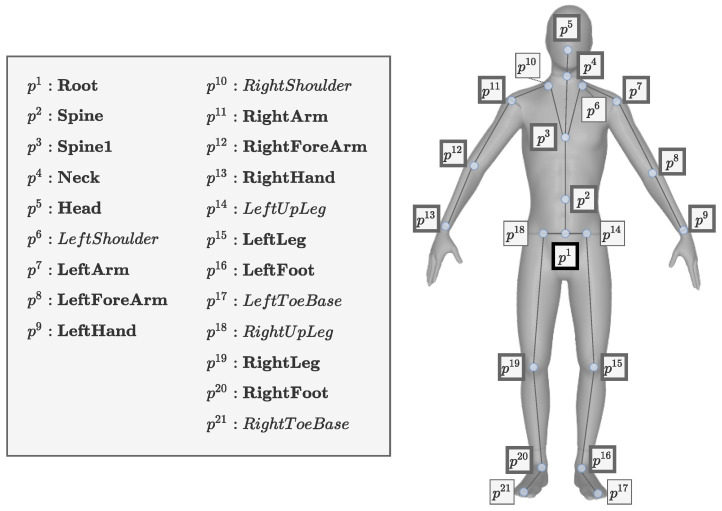
Skeleton structure: the skeleton for our study had 21 joints ($p^{1}$ to $p^{21}$). The 15 joints that were used to evaluate the errors in the experiments are highlighted in bold and thickly lined boxes. The joint list for evaluating the errors was as follows: Hip, Spine, Spine1, Neck, Head, LeftArm, LeftForeArm, LeftHand, RightArm, RightForeArm, RightHand, LeftLeg, LeftFoot, RightLeg, and RightFoot.

**Figure 5 sensors-22-04846-f005:**
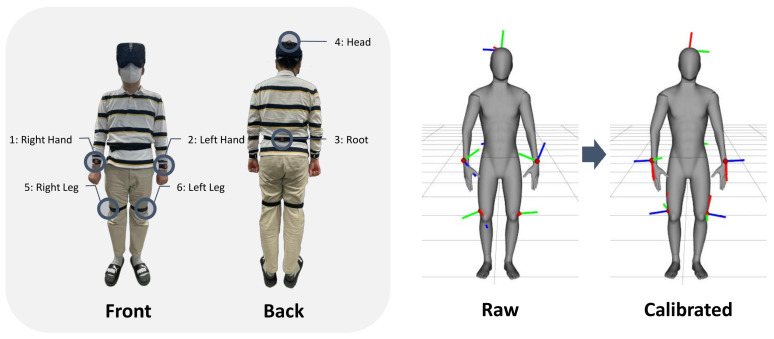
The IMU placement and calibration: (**Left**) the placement of the six IMUs that were attached to the body: right hand, left hand, pelvis (root), head, right leg, and left leg; (**Right**) the results of the IMU sensor calibration to the body-centric coordinates.

**Figure 6 sensors-22-04846-f006:**
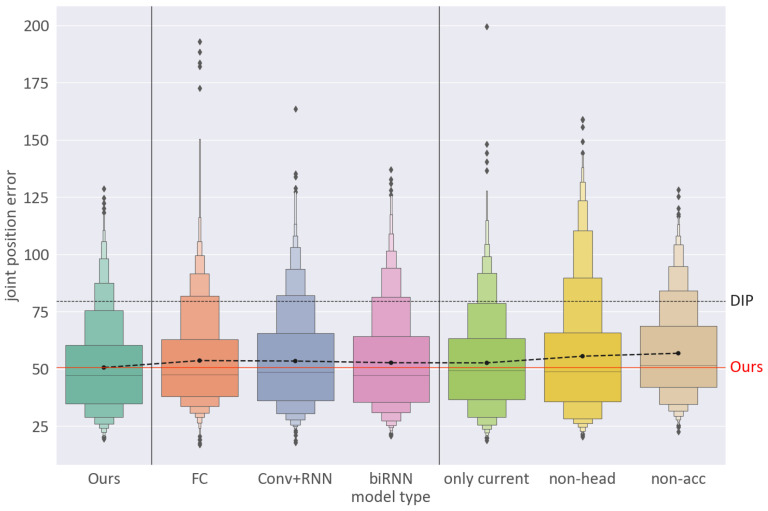
An evaluation of the mean per joint position error. The graph shown here is based on Table 1. We recorded the errors in each frame and showed that our network achieved a better performance than the other variants and DIP [15].

**Figure 7 sensors-22-04846-f007:**
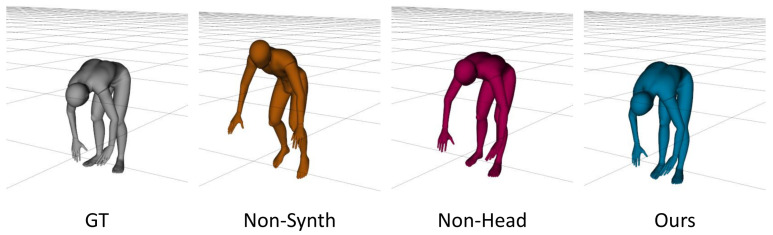
A comparison of the 3D model results. The figure shows the ground-truth and the real-time results of three types of models: the model that was trained excluding synthetic data (Non-Synth); the model that was trained excluding head height data (Non-head); and our model.

**Figure 8 sensors-22-04846-f008:**
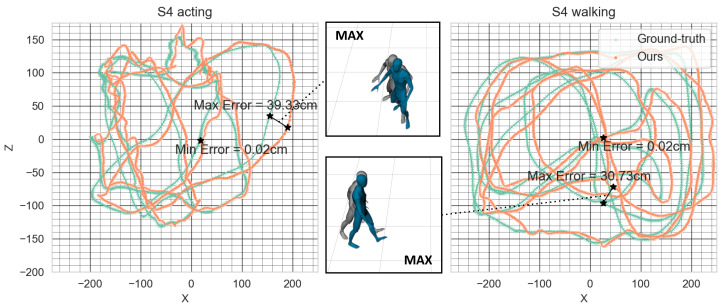
The trajectory of S4 (Subject 4) from the TotalCapture dataset. The overlapping frames in the middle of figure show the maximum value of the location tracking error in each motion sequence.

**Figure 9 sensors-22-04846-f009:**
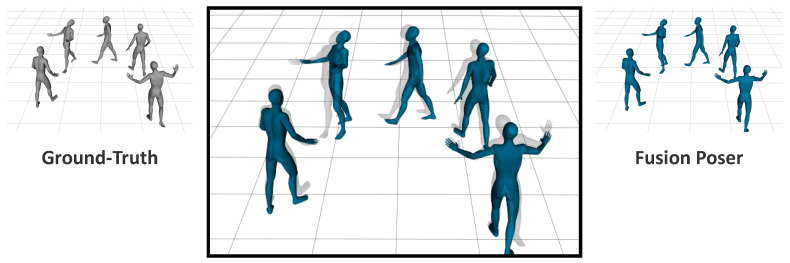
A comparison between the predicted and ground-truth poses using TotalCapture data (S4). By overlaying the predicted and ground-truth results, we could visualize the differences.

**Figure 10 sensors-22-04846-f010:**
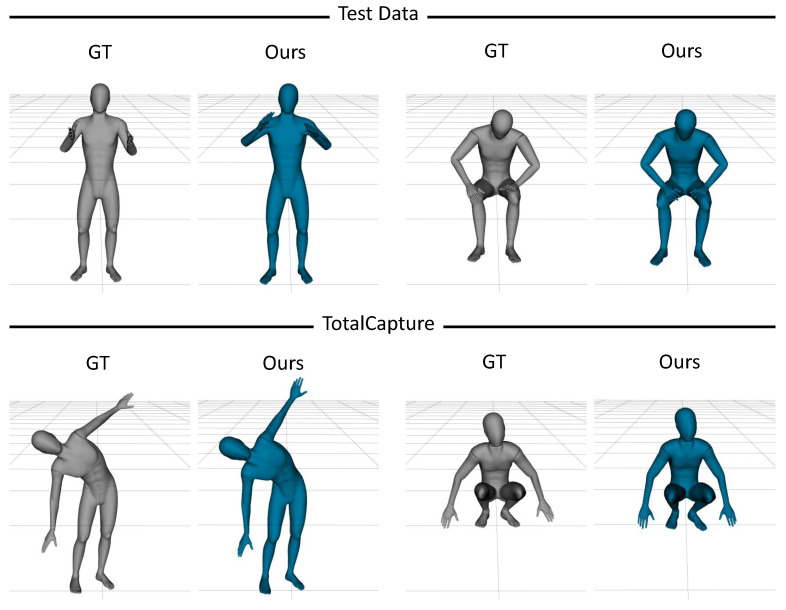
Example results using our mocap data and TotalCapture data.

**Figure 11 sensors-22-04846-f011:**
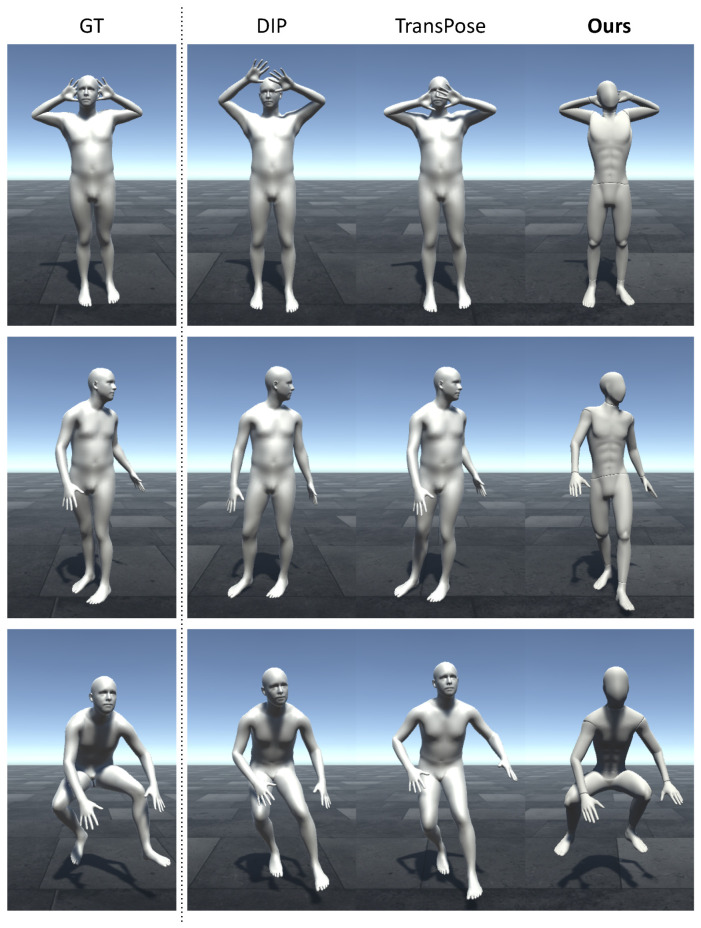
A qualitative comparison between the online pose estimation results of our method and those of previous works: the first column shows the ground-truth of the selected frame from the TotalCapture dataset, then the reconstructed SMPL poses from the estimation results of DIP are in the second column and those of TransPose are in the third column.

**Figure 12 sensors-22-04846-f012:**
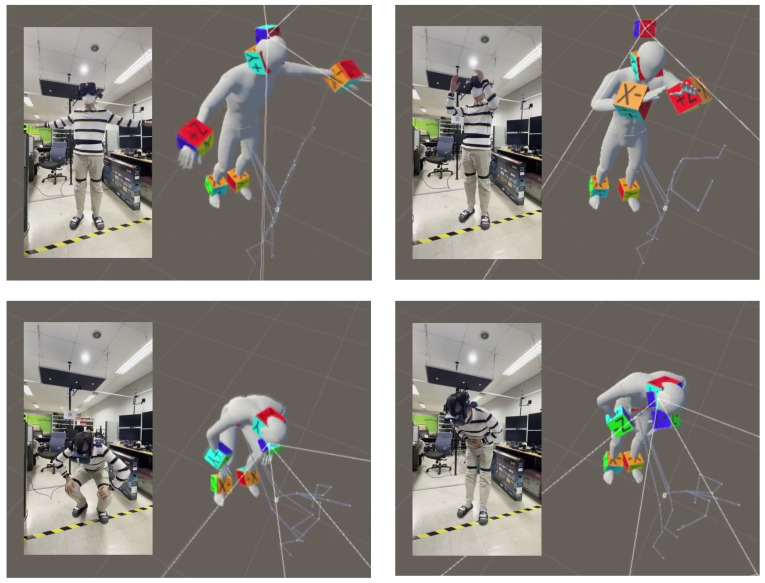
The real-time application using sample frames in Unity 3D. Our implemented application took the HMD sensor as input and reconstructed the 3D modeling body after a slight delay.

**Figure 13 sensors-22-04846-f013:**
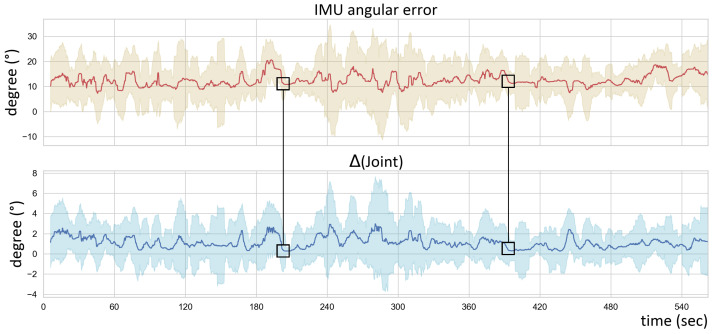
The time-dependent changes in the angular errors of the inertial sensors and the angles of the joints to which the sensors are attached. The errors represent the angle differences between the IMU measurement/synthesized data and the motion capture data within same frame. The errors increased with the continual movement of the sensors and returned to the initial error value when the movement stopped.

**Figure 14 sensors-22-04846-f014:**
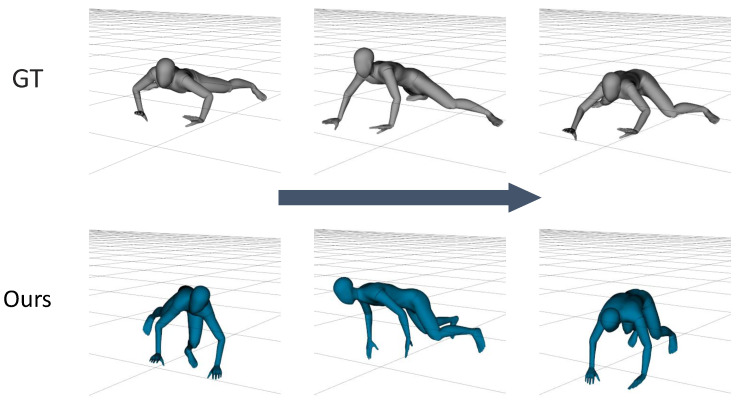
Examples of failure cases for the crawling motion in the TotalCapture dataset: (**top**) the ground-truth motion; (**bottom**) our predicted motion. It was a continuous motion with 15 frame intervals.

**Figure 15 sensors-22-04846-f015:**
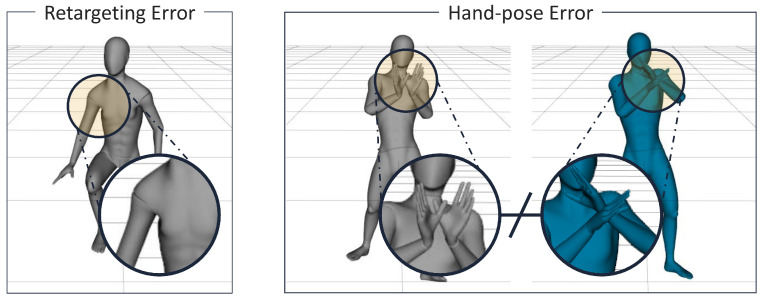
Examples of failure cases for retargeting the body and estimating the hand pose: (**Left**) retargeting error; (**Right**) incorrectly estimated hand pose. This is an example of an incorrect prediction during the motion of putting the hands together.

**Table 1 sensors-22-04846-t001:** The evaluation of our pose estimation network using different input variables and network architecture variants with the TotalCapture [18] dataset. The errors of each model are described as the mean ($\mu_{pos}$) and standard deviation ($\sigma_{pos}$) of the joint position error in millimeters and the mean ($\mu_{ang}$) and standard deviation ($\sigma_{ang}$) of the joint angle error in degrees (°).

	$\mu_{\mathrm{pos}}$ (mm)	$\sigma_{\mathrm{pos}}$ (mm)	$\mu_{\mathrm{ang}}$ (°)	$\sigma_{\mathrm{ang}}$ (°)
**Ours**	**50.51**	20.07	11.31	4.58
DIP	79.42	32.15	13.67	9.59
TransPose	68.51	41.43	12.93	6.15
FC (512)	53.54	21.58	**10.43**	4.36
Conv+RNN	53.38	22.08	11.13	4.72
biRNN	52.60	21.19	10.99	4.44
only current	52.55	21.44	11.51	4.56
non-head	55.45	27.19	11.31	4.87
non-acc	56.74	20.47	11.61	5.06

## Data Availability

Publicly available datasets were analyzed in this study. These data can be found here: https://github.com/LuzyCat/FusionPoser (accessed on 20 June 2022). This paper used third party data to train and evaluate the model. Restrictions apply to the availability of these data. The training data were obtained from the CMU Graphics Lab Motion Capture Database and are openly available from http://mocap.cs.cmu.edu/ (accessed on 14 April 2022). The training data were also obtained from the TotalCapture dataset and are available from https://cvssp.org/data/totalcapture/data/ (accessed on 14 April 2022) with the permission of Andrew Gilbert.

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
