# Peer review of "Fusion Poser: 3D Human Pose Estimation Using Sparse IMUs and Head Trackers in Real Time"

_sensors, 2022, doi:10.3390/s22134846_

Round 1

Reviewer 1 Report

The paper presents fusion work of 3D Human Pose Estimation using Sparse IMUs and Head Tracker in real-time. The work is interesting and the structure is ok. The novelty is given with detail and well structured. The overal paper is ok. There are some suggestions. In experiments, more different challenge pose should be involved for comparison. In addition, it is interesting to see how it works when reduce the sensors , e.g. only on the wrist part which most application based on here. 

Reviewer 2 Report

Dear Authors,

the topic of the manuscript itself is interesting and presents an important research question. Moreover, the manuscript itself is extensive and shows significant effort conducted. 

However, the study fails is some crucial interpretation of the measured values, the method described is not scientifically sound and the manuscript itself is poor in presentation, style, and structure.

In more detail, my concerns, comments, and suggestions are (stated in order of appearance in the manuscript) as follows. 

All abbreviations used in the abstract should be defined in full (e.g., HMD). 

All abbreviations used in the manuscript text should be defined in full even if already used and defined in the abstract (e.g., VFX, VR, DIP, SIP, …). 

You state that the error in joint angle or distance must be at a level that can recognize the action. Unfortunately, you do not clarify what this means for your study and for the obtained results. This criteria is not considered later on in the manuscript and most of the results are qualitative and almost subjective in nature as oppose to being quantitative.  

In Lines 193-195 you state that “In addition, our approach uses another type of sensor, which is a global head pose tracker that can be any device such as an HMD tracker or motion capture device.” – Since this should be a scientific article, you should be clear regarding which device you used and to which device the presented results refer to. Additionally, you can mention that any similar device can be used for the task. 

I assume that by “the orientation and acceleration data sequence” in Line 73 (and elsewhere) you mean angular velocity and acceleration signals? 

Do you feed the timestamps of signal samples to the model as well? Sensors do not necessarily provide for uniform sampling. Did you check?

Spatio-temporal learning in line 74 probably should not be capitalized.

Sentence in Lines 75-76 is repeated immediately after.

What is meant by height sequence in Line 76?

Is ‘frame’ used for time sample?

Most of the Lines 68-108 would better fit the Method section instead of the introduction.

‘in-the-wild’ in Line 139 could and should be replaced with ‘out-of-lab’. Or you actually indented to track pose in some wilderness?

Lines 199-207 need style, language, and content per se editing. As does much of the other text.

Angular velocity data depends on the initial and final position/orientation of the segment captured. How were the tests performed – did the subject always start from the same position? When coming to a pose from each different pose/orientation, the angular velocity data will be significantly different. Therefore, using angular velocity data as presented in this manuscript, to estimate a pose is faulty. Have you tested the accuracy of the models using only acceleration data?

How is the choice of the NN structure (BiRNN) argumented? Since BiRNN depends on past and future data, how does this affect real-time pose estimation?

In Lines 159-160 you state that: “Reconstructing the human pose from sparse IMUs with high accuracy is a challenging problem because the data from sensors is insufficient to configure the human pose.” However, this is only partially true. The main difficulty occurs even when estimating one segment position, due to sensor inaccuracies (especially bias drift). And this is why many studies include sensor fusion models, as stated later on in Lines 161-162. Additionally, this is why the errors in pose estimation will be ever larger, the longer the test lasts. The duration of your tests is not even mentioned, let alone considered.  

In Lines 182-183 you state that you combine IMU and head tracker data to obtain global position. What is meant with this – position in the global coordinate system or a one-point position of the human body (e.g. the center of mass)?

How do you calculate the orientation quaternion from the angular velocity data? There is no angular velocity occurrence in Equation (3).

In the title of Figure 1, “This figure describes” should be removed. This is an example of bad practice and is usually avoided in scientific.

What is meant by “root join” in Section 3.2.1? How is Rroot in Equation (5) obtained? In general, the text in Lines 226-234 is poorly written.

In lines 336-338 you state that the IMU measures in the local magnetic fields. This sentence has no sense. The IMU measures in its local coordinate system, defined by how the sensitivity axis themselves are placed during the manufacturing process. In addition, acceleration and angular velocity, which you are using, are measured completely independently of the magnetic fields. 

Regarding the results section, when compared to other methods, as presented in Table 1, the proposed method shows no significant improvement in performance.  Additionally, reported maximum errors 30-40 cm alone indicate that the proposed solution fails for the problem at hand and that a different approach altogether should be implemented (which would also focus on sensor fusion, sensor inaccuracies and the correct use of the measured angular velocity values).

Qualitative analysis results presented in Figures 9 and 10 have no scientific significance – the differences depicted could be a one case occurrence. The “remarkable quality improvement” is a huge exaggeration and is definitely not supported by the data presented in Table 1.

Round 2

Reviewer 1 Report

The revised paper is ok for publish

Author Response

Dear Reviewer,

We thank for you carefully reviewing our manuscript and giving us many valuable comments.

Reviewer 2 Report

This reviewer acknowledges that significant changes have been made to the original manuscript. 

Especially, this reviewer acknowledges that what is actually used from the IMU is now stated in a clearer way. However, if you orientation, already provided by the IMU, is used, then the authors should elaborate what sensor fusion algorithms are used and how accurate is the obtained orientation, especially with respect to the duration of measurement. This has far reaching consequences on the results of the predictions.

And if indeed the already estimated angular orientation is used, what is the point of additionally using acceleration as the prediction model input? How do the authors argument this redundancy? What are the improvements in the results if both, orientation and acceleration are used, as opposed to using only orientation. The authors aim at providing real-time predictions. Any unnecessary complexities in the implemented model in this context should hence be avoided.
